# Rationale and Design of a Randomized Controlled Clinical Trial on the Safety and Efficacy of Flecainide versus Amiodarone in the Cardioversion of Atrial Fibrillation at the Emergency Department in Patients with Coronary Artery Disease (FLECA-ED)

**DOI:** 10.3390/jcm12123961

**Published:** 2023-06-10

**Authors:** Panagiotis Tsioufis, Dimitris Tsiachris, Ioannis Doundoulakis, Athanasios Kordalis, Christos-Konstantinos Antoniou, Panayotis K. Vlachakis, Panagiotis Theofilis, Eleni Manta, Konstantinos A. Gatzoulis, John Parissis, Konstantinos Tsioufis

**Affiliations:** 1First Department of Cardiology, Hippocration General Hospital, National and Kapodistrian University of Athens, 11527 Athens, Greece; ptsioufis@gmail.com (P.T.); doudougiannis@gmail.com (I.D.); akordalis@gmail.com (A.K.); elenmanta@gmail.com (E.M.); kgatzoul@med.uoa.gr (K.A.G.); ktsioufis@hippocratio.gr (K.T.); 2Athens Heart Center, Athens Medical Center, 11526 Athens, Greece; 3Emergency Department, Attikon University Hospital, National and Kapodistrian University of Athens, 12462 Athens, Greece; jparissis@yahoo.com

**Keywords:** atrial fibrillation, emergency department, flecainide, cardioversion, amiodarone

## Abstract

Pharmacologic cardioversion is a well-established alternative to electric cardioversion for hemodynamically stable patients, as it skips the risks associated with anesthesia. A recent network meta-analysis identifies the most effective antiarrhythmics for pharmacologic cardioversion with flecainide exhibiting a more efficacious and safer profile towards faster cardioversion. Moreover, the meta-analysis of class Ic antiarrhythmics revealed an absence of adverse events when used for pharmacologic cardioversion of AF in the ED, including patients with structural heart disease. The primary goals of this clinical trial are to prove the superiority of flecainide over amiodarone in the successful cardioversion of paroxysmal atrial fibrillation in the Emergency Department and to prove that the safety of flecainide is non-inferior to amiodarone in patients with coronary artery disease without residual ischemia, and an ejection fraction over 35%. The secondary goals of this study are to prove the superiority of flecainide over amiodarone in the reduction in hospitalizations from the Emergency Department due to atrial fibrillation in the time taken to achieve cardioversion, and in the reduction in the need to conduct electrical cardioversion.

## 1. Introduction

Atrial fibrillation (AF) is the most frequent heart arrhythmia in the adult population [1]. This is reflected in the number of ED visits with a primary diagnosis of AF, as well as in the high percentage of admissions due to AF [2,3]. According to the current data, the annual number of visits to the Emergency Department due to atrial fibrillation continues to rise [4]. Hospital admissions related to atrial fibrillation also tend to increase, increasing the financial burden to the national health system [5]. It has been proven that interventional programs at the ED can reduce admissions due to AF and the associated costs [6,7,8]. In this setting, prompt cardioversion of atrial fibrillation at the Emergency Department has been independently proven to significantly reduce admissions and the related costs [9], as well as its progression to persistent atrial fibrillation [10], and the serious cardiovascular risks associated with it [11]. Data from Italy [12] and the USA [4] show that the percentage of patients coming to the ED and subsequently becoming hospitalized is much higher than the percentage of patients coming to the ED for any diagnosis and then hospitalized. Furthermore, it is noteworthy that the majority of these patients that visited the ED with AF had a calculated CHA2DS2-VASc score ≥ 2 and were already under optimal anticoagulation. Therefore, these patients could theoretically be submitted to cardioversion at the Emergency Department and could possibly be discharged.

Pharmacologic cardioversion is a well-established alternative to electric cardioversion for hemodynamically stable patients as it skips the risks associated with anesthesia [13,14]. However, the current guidelines do not prioritize between antiarrhythmic agents and do not take into account the time taken to restore the sinus rhythm [13,14,15].

Antiarrhythmic drugs, such as amiodarone, flecainide and propafenone, are considered as the primary choice for hemodynamically stable patients. In patients without structural heart disease, the use of intravenous flecainide or propafenone is recommended, while for patients with a history of heart failure or SHD, amiodarone is the drug of choice. A recent network meta-analysis identifies the most effective antiarrhythmics for pharmacologic cardioversion with flecainide exhibiting a more efficacious and safer profile towards faster cardioversion. Moreover, the meta-analysis of class Ic antiarrhythmics revealed an absence of adverse events when used for pharmacologic cardioversion of AF in the ED, including patients with structural heart disease. 

Over the last years, the importance of ‘time-consuming’ pharmacological cardioversion of AF was depreciated in favor of electrical cardioversion. ED cardioversion as a management approach for stable patients with recent-onset AF potentially allows for discharge directly from the ED, and, thereby, may be advantageous towards health care resource allocation, cost, and potential patient satisfaction perspectives. We hypothesize that using flecainide for the cardioversion of eligible patients will allow for faster cardioversion times, while maintaining the same or a better safety profile than amiodarone.

## 2. Objectives

The primary goals of this clinical trial are to prove the superiority of flecainide over amiodarone in the successful cardioversion of paroxysmal atrial fibrillation at the Emergency Department and to prove that the safety of flecainide is non-inferior to amiodarone in patients with coronary artery disease without residual ischemia and an ejection fraction over 35%. The secondary goals of the study are to prove the superiority of flecainide over amiodarone in the reduction in hospitalizations from the Emergency Department due to atrial fibrillation in the time taken to achieve cardioversion, and in the reduction in the need to conduct electrical cardioversion. A successful cardioversion is defined as the restoration of the sinus rhythm, documented by a 12 lead ECG lasting at least 1 h. 

## 3. Materials and Methods

### 3.1. Trial Design Overview 

The FLECA-ED study [NCT05549752] is an ongoing, randomized, controlled, multi-center trial that aims to fill in the gaps relating to the use of flecainide for the cardioversion of AF in patients with a history of coronary artery disease without residual ischemia and a left ventricular ejection fraction > 35%. The trial’s start date was 24 March 2023, and the expected completion date is 1 November 2024. 

The trial population will be 200 consecutive all-comers to the emergency department with paroxysmal atrial fibrillation. These patients will be randomly assigned to the amiodarone or flecainide group, in a head-to-head comparison of the effectiveness and safety profile of the two drugs in this specific population. The study entails a baseline visit followed by the intervention visit (both at the ED), as well as one day follow-up at the investigation site for re-evaluation and a 30-day remote follow-up (phone visit). All eligible patients arriving at the ED that sign the informed consent and are enrolled in the trial will be fitted with a 24 h ECG halter device prior to drug administration in order to assess the incidence of PVCs, NSVT, SVT and bradycardia, as well as the time taken for cardioversion. For 6 h following the initiation of the drug administration, all the enrolled patients will stay at the Emergency Department under close monitoring. If the patient’s rhythm is not successfully restored to a sinus rhythm at 6 h, the patient will be admitted to the hospital and will receive intravenous amiodarone. 

This study will provide substantial data for the safety of flecainide in patients with coronary artery disease without residual ischemia. Furthermore, the direct comparison to amiodarone will demonstrate flecainide’s performance in terms of frequency and the time needed for successful cardioversion.

The FLECA-ED trial is being conducted in Emergency Departments in three large tertiary hospitals in Athens, Greece. Two hospitals (‘Hippokration’ and ‘Attikon’ General Hospitals) are public hospitals, while the third (Athens Heart Center) provides care only to privately insured patients. Each of these hospitals has a large daily influx of patients and a significant experience in research work. The trial is funded by WinMedica (Athens, Greece). All the participants in this study provide written, informed consent. All study personnel have training and certification in human subjects’ research. The whole study protocol is approved by the Hospitals’ Institutional Review Board, the National Organization for Medicines and National Ethical Committee. The authors take sole responsibility for the design and conduct of this trial, the analyses and the composition of this design paper and its contents.

### 3.2. Eligibility Criteria

Patients are eligible if the duration of paroxysmal atrial fibrillation is less than 48 h at the time of cardioversion or less than 7 days with usage of NOACs for more than 30 days at the time of cardioversion. Atrial fibrillation will be confirmed with a standard 12 lead electrocardiogram (ECG). Patients will be included only if they meet strict criteria that ensures that residual ischemia is not present. More specifically, only patients with a 1 year history of PCI or 3 year history of CABG will be included. If this criterion is not met, the patients must have a negative imaging-based stress test within the previous year. Patients exhibiting angina or similar symptoms at the ED will be excluded, as is the case for patients with a history of acute coronary syndrome within the last year. Patients are eligible regardless of their EHRA symptom classification score. Patients with a history of thyroid dysfunction are not excluded. Table 1 lists the inclusion and exclusion criteria for the trial enrolment.

### 3.3. Randomization

Randomization is performed via an eCRF platform developed by PHARMASSIST CRO (Athens, Greece), which is also used to document all the study contacts, assessments and outcomes. 

Patients and researchers are aware of the treatment group that patients have been randomized to. The two antiarrhythmic drugs, amiodarone and flecainide, have different dosages and a different time and method of administration, making the design of a single or double-blinded trial design difficult in an Emergency Department setting. 

The process of active randomization in each of the two groups in a 1:1 ratio contributes to the prevention of selection bias and is undertaken by an independent authorized researcher who is not taking part in the rest of the trial’s procedures.

### 3.4. Sample Size

In our meta-analysis, we estimated the pooled cardioversion rate by flecainide within 4 h 65% whereas by amiodarone 23% (17). Taking 65% as our point estimate, the steering committee of FLECA-ED considers that flecainide can be regarded as non-inferior to amiodarone in patients with ischemic cardiomyopathy if the lower margin of the confidence interval does not go lower than the pooled rate (65%) by more than 20% (i.e., if it is not lower than 45%). Based on these assumptions, we expect that a sample size of 96 patients per arm is required to achieve 80% power to demonstrate the efficacy of flecainide in these patients, when a type I error rate is set at 0.05. Given the close follow-up of our patients in our registry, we do not expect a substantial dropout rate and thus aim to recruit a total of 200 (100 per arm) patients.

### 3.5. Study Phases and Assessments

Table 2 summarizes the type and frequency of safety and efficacy outcomes to be measured at each patient’s assessment by the investigators or designated site staff. 

Trial design with details on assessments per each phase are briefly outlined below:

#### 3.5.1. Screening Phase (Vscr, Emergency Department)

In Vscr, the site staff will assess eligibility according to the inclusion and exclusion criteria and obtain written informed consent. The informed consent and the whole study protocol is approved by the National Ethics Committee. Patients signing the informed consent will also explicitly consent to telephone calls by the investigators for follow-ups. A dated copy of the signed informed consent will be given to the patient. The enrolment of the participants will be considered valid once they have given their consent, completed the screening process and are deemed eligible. The moment of inclusion is defined as when the patient signs and dates the informed consent form. Any assessments and examinations related to the study are conducted after the informed consent has been obtained. Patients who sign the informed consent form also explicitly consent to telephone calls from the investigators for ongoing follow-ups.

At the ED, all patients are under continuous ECG monitoring. Once the signed informed consent has been obtained, all patients are fitted with a 24 h ECG holter monitoring device, without a removal of the ED ECG monitor. The flowchart of a patient’s baseline assessments is shown at Figure 1.

#### 3.5.2. Intervention Visit (V0)

The intervention visit is defined as the time at which the study staff intravenously administers the assigned drug and coincides with the screening visit. The drug that will be administered to each patient will be determined randomly, through a simple randomization process, as detailed in the relevant section. Patients in both groups will remain at the ED for a total of 6 h after starting the treatment, regardless of cardioversion status. If no adverse events occur during this time, including an AV block, sinus bradycardia < 50 bpm, syncope, presyncope, or systolic blood pressure < 90 mmHg and a cardioversion to sinus rhythm is successful, the patient will be discharged. Otherwise, the patient will be admitted to the hospital. Patients receiving flecainide with an unsuccessful cardioversion at 6 h will be hospitalized and receive amiodarone. Patients with an unsuccessful drug cardioversion after 24 h will undergo electrical cardioversion, according to the clinical judgment of the attending physician. All patients (i.e., those who were discharged at 6 h following a successful cardioversion and those who were hospitalized) will be under continuous 24 h ECG halter monitoring. The hospitalized patients will also be under continuous ECG monitoring until the time of a cardioversion. The 24 h ECG halter will be removed at 24 h from drug administration initiation, regardless of the cardioversion status.

In case of hemodynamic instability during or after the administration of the drug, or if the Δ (troponin at 1 h) is above the limit of the hospital’s laboratory troponin cut-off levels, the administration of the drug will cease, and the patient will be treated according to the guidelines. At discharge, the administration or non-administration of other medication will follow the guidelines, with the exception that patients will not receive treatment with a class Ic antiarrhythmic agent. 

#### 3.5.3. Follow-Up Phase (from Intervention Visit to 1 Month)

The follow-up of the enrolled subjects will commence from the intervention visit and will extend to 30 days at the common study termination date.

At 24 h, patients will visit the investigation site for a clinical evaluation, including a physical examination, ECG and echocardiography, the removal of the 24 h ECG halter device and an adverse events assessment. 

At 30 days, a phone-driven follow-up evaluation is conducted for the assessment of any of the study’s outcomes and adverse events (AE). Patients will also be instructed to report earlier to the study center if they have symptoms suggestive of an AE. 

### 3.6. Interventions

Individuals randomized to the flecainide arm will intravenously receive flecainide at a dose of 2.0 mg/kg (maximum dose of 150 mg) throughout the 10 min in the emergency department. Flecainide will be dissolved in a 100 mL D/W 5% solution.

Patients randomized to the amiodarone arm will intravenously receive amiodarone at a dose of 5.0–7.0 mg/kg over 1 h in the emergency department, as well as a follow-up dose of 50 mg/h (a maximum dose of 1000 mg) for up to 24 h (Figure 2). 

Intravenous metoprolol is administered in patients with atrial fibrillation exhibiting a ventricular rate over 100 bpm, according to a physician’s decision. 

Hemodynamically stable patients with an unsuccessful pharmacological cardioversion at 24 h will undergo electrical cardioversion, according to the clinical judgement of the treating physician.

Any prior use of an antiarrhythmic agent will be documented in the case report form (CRF) and all concomitant medications or other therapies will also be documented. For patients withdrawing from the study, the reasons and any follow-on therapy are to be documented in the CRF.

### 3.7. Outcomes

The study’s primary endpoints are the frequencies of successful cardioversion within 6 h of drug administration as well as the combined frequency of PVCs, NSVT, SVT, bradycardia and hypotension during the same time frame. The main secondary endpoints are the frequency of patient discharges from the Emergency Department with a sinus rhythm and the frequency of successful cardioversion to a sinus rhythm within 24 h. All adverse effects, including extracardiac, will be assessed and categorized based on their severity, type and frequency. Table 3 lists all the study primary and secondary endpoints.

### 3.8. Statistical Analysis

All safety and efficacy results will be summarized using descriptive statistics.

The continuous variables will be expressed as mean ± standard deviation or median and interquartile range ((IQR), 25th, 75th percentile) depending on the normality of the distribution. The categorical values will be presented as frequencies and percentages. 

All primary and secondary endpoints will be analyzed according to the “intention to treat” principle, i.e., according to the treatment given, without reference to the treatment compliance or changes. Additional per-protocol sensitivity analyses will be performed.

No statistical adjustment will be performed for multiple secondary endpoints in their analysis, but reporting of all the secondary endpoint analyses will make it clear whether the primary endpoint was statistically significant and will report the number of secondary endpoints proposed as a priori in the study protocol.

A safety assessment will be based on the adverse event reports collected between the signing of the informed consent form and the completion of the final follow-up visit for each participant.

In case of an interim analysis, the Pocock method will be used and a *p*-value of 0.03 for the interim and 0.03 for the final analysis will be used. The data will be analyzed according to the “intention to treat” principle.

## 4. Discussion

Here we report the rationale and design for a multi-center randomized and controlled clinical trial, which aims to demonstrate the safety and efficacy of flecainide in the prompt cardioversion of atrial fibrillation in patients with coronary artery disease and EF > 35% without residual ischemia in an ED setting. We selected amiodarone as the relational agent due to its wide adoption and known safety and efficacy profile. Vernakalant was considered as well, but its limited availability and higher cost made the comparison not feasible at this time. 

In patients without structural heart disease, the use of intravenous flecainide or propafenone is recommended, while for patients with a history of heart failure or SHD, amiodarone is the drug of choice. Vernakalant is a novel agent with a good safety profile and performance in terms of successful cardioversion in a timely fashion. In the recent post-hoc analysis of the large, international, observational SPECTRUM study [16], IV vernakalant was well-tolerated, albeit with a few (0.9%) adverse events involving significant bradycardia, including sinus arrest. Successful cardioversion was reported in 67.8% of episodes and the median length of hospital stays at the ED was 7.4 h. Vernakalant may be used in patients with a history of coronary artery disease, which is a patient group where amiodarone is typically used. However, vernakalant’s limited availability and debatable cost-effectiveness could hinder its wide adoption at the present time and should be avoided in cases of severe heart failure, hypotension or acute coronary syndrome. 

On the other hand, class IC agents, such as flecainide, are contraindicated for the cardioversion of patients with coronary artery disease post-revascularization, as well as patients with ischemic cardiomyopathy and preserved ejection fraction, although there is a literature gap in this setting [13,17,18]. At the same time, the effectiveness and safety of flecainide for the restoration of a sinus rhythm has been proven in a series of clinical trials, some of which being randomized, as well as in several meta-analyses [17,19,20]. Flecainide has been associated with a shorter cardioversion time when compared to amiodarone or propafenone, which could pave the way for early hospital discharge directly from the ED.

The main risks related to antiarrhythmic class Ic agents, including flecainide, are the negative inotropic and proarrhythmic effects. The recommendation to avoid these agents in patients with atrial fibrillation and structural heart disease with or without ischemic coronary artery disease has been based solely on the results of the CAST study performed in 1991 [21]. That study aimed to prove that the suppression of ventricular ectopy using flecainide after a myocardial infarction could lead to a reduction in the incidence of sudden cardiac death. The population was comprised of patients with a history of MI within 2 years, a reduced left ventricular ejection fraction and frequent premature ventricular contractions. The patients were segregated into three groups, where flecainide, encainide or morisizine was administered, respectively, throughout the course of the trial. However, the study faced an early termination at 10 months due to an increased mortality in patients receiving class IC agents. 

This trial, however, had several shortcomings: patients with revascularized coronary artery disease were excluded and there were no checks performed prior to the inclusion for residual ischemia or angina. Furthermore, flecainide was administered for a prolonged timespan, and not for the short time needed for the cardioversion of AF. The subsequent sub-analyses of the data of the CAST study showed that patients with angina or electric instability were more likely to have worse outcomes upon flecainide administration [22,23]. This observation could perhaps uncover the interaction between the drug and the ischemic foci as the reason for the increased mortality in these patients. It is noteworthy that the population of these sub-analyses had a non-Q myocardial infarction, which could potentially be the basis for the rejection of the hypothesis that the interaction of flecainide with the scarred substrate caused the adverse events.

The European Rythmol/Rythmonorm Atrial Fibrillation Trial (ERAFT) that followed studied the safety and efficacy of propafenon in the prevention of symptoms of paroxysmal atrial fibrillation (PAF) in patients with a history of myocardial infarction but without unstable angina nor history of ventricular arrhythmias [24]. The results were encouraging, as no increase in mortality and a frequency of ventricular arrhythmogenesis were observed during the administration of these class IC antiarrhythmic agents. 

In our recent meta-analysis [17], 473 patients with ischemic cardiomyopathy or heart failure were included and the effectiveness and safety profile of the class IC agents for the cardioversion of PAF was examined. The results showed that intravenously administered flecainide is the most effective choice for the pharmacologic cardioversion of atrial fibrillation, as only two patients needed to be treated in order to cardiovert one within 4 h. Only one patient exhibited ventricular arrhythmia in a 1992 study, 6% of patients demonstrated hypotension and 0.9% bradycardia. In studies that followed (prior to 2004), and are even referenced in the latest ESC guidelines, no episode of ventricular tachycardia was recorded. Similar results were found at the propafenone group. These findings could demonstrate that ischemia, and not the presence of structural heart disease, may be the culprit for the increased risk associated with class IC agents for the cardioversion of AF. 

In a recent retrospective study, 3445 patients receiving a class IC agent were compared to 2216 patients under sotalol or dofetilide, in a time span of 16 years [25]. The patients had variable degrees of coronary artery disease but no history of ventricular tachycardia, ICD or non-revascularized myocardial infarction. The authors found that IC agents are not associated with an increased mortality in certain patient groups that these drugs are currently restricted in, such as select patients with nonobstructive CAD without a history of VT. 

While designing the trial, patient safety was our utmost priority, and we incorporated a slew of strategies to minimize the risk for patients. First, this is evidenced by the stricter implementation of the ESC guidelines regarding residual ischemia. More specifically, in case of a CABG procedure, we required the date of the surgery to be within the last 3 years instead of 5 years. Secondly, we excluded all patients with suspicious symptoms resembling an acute coronary syndrome and we enrolled patients with a negative high-sensitive troponin result. As an extra layer of precaution, following enrolment and intervention initiation, all patients without suspicious symptoms were assessed using a second measurement of high-sensitivity troponin at 1 h. 

Furthermore, we undertook the decision to place a 24 h ECG halter device monitor, on top of the 6 h ECG monitoring at the ED, to ensure the timely detection of any adverse event that would otherwise be missed, such as PVCs, NSVT, SVT or bradycardia. As for the decision to discharge hemodynamically stable patients that have been cardioversed at 6 h, this was based on the recommendations by Dan et al. that continuous medical supervision and ECG monitoring during drug infusion and afterwards for at least half of the drug half-life is needed. Based on this, we opted to not monitor the discharged patients by other means, except from the halter device. These measures were deemed as acceptable by the National Organization for Medicines and National Ethical Committee during the approval process of this trial. 

## 5. Conclusions

We believe that the embodiment of currently restricted but more efficacious drugs in the clinical workflow for the treatment of patients with coronary artery disease, such as flecainide, could satisfy the need to reduce the length of stay at the ED and hospitalizations. Our trial aims to reappraise the role of flecainide in the cardioversion of hemodynamically stable patients, presenting to the ED without residual ischemia and an EF > 35%. Depending on the trial’s results, the trial could be extended to include all asymptomatic patients presenting to the ED with an unknown medical history, following a bedside examination and ultrasonography to exclude severe cardiac anomalies.

### Current Status

Protocol version 1.0 (27 July 2022); enrolment (recruitment) start date: 24 March 2023; projected enrolment (recruitment) completion date: 1 November 2024. Approval was granted by the National Organization for Medicines, National Ethical Committee as well as by the hospitals’ institutional review boards (IRB).

## Figures and Tables

**Figure 1 jcm-12-03961-f001:**
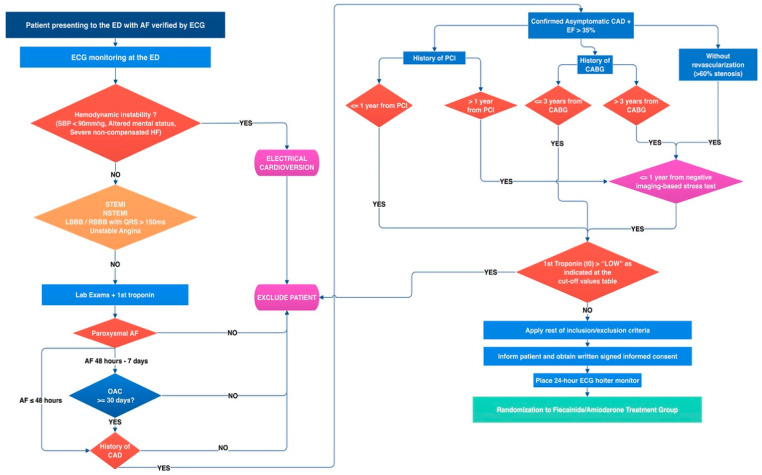
Screening phase flowchart.

**Figure 2 jcm-12-03961-f002:**
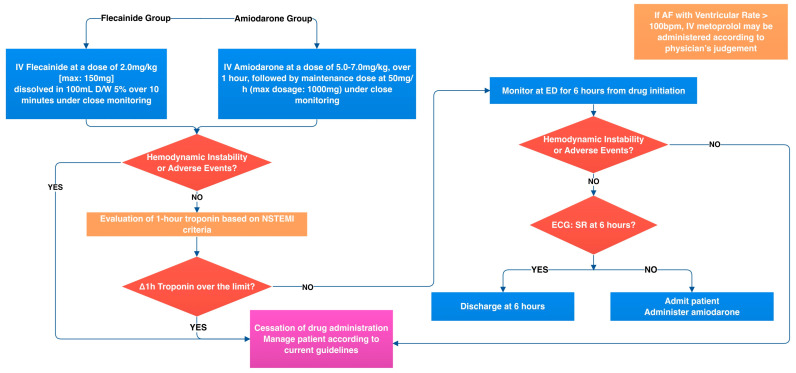
Intervention visit flowchart.

**Table 1 jcm-12-03961-t001:** Eligibility criteria.

Inclusion Criteria
1. Age: 18–85 years old 2. Paroxysmal Atrial Fibrillation, documented by 12-lead ECG, with one of the following: a. Atrial Fibrillation onset less than 48 h from the time of presentation to the Emergency Department b. Atrial Fibrillation onset between 48 h and 7 days from the time of presentation to the Emergency Department and if patient has been on anticoagulation for at least 30 days c. History of Coronary Artery Disease without residual ischemia, defined by one of the following criteria: ▪ PCI ≤ 1 year, or ▪ CABG ≤ 3 years, or ▪ Negative imaging-based stress testing within 1 year, and: ▪ History of known coronary artery stenosis > 60% without revascularization, or ▪ PCI ≥ 1 year, or ▪ CABG ≥ 3 years 3. Ejection Fraction > 35% (documented by cardiac ultrasound at the Emergency Department, or within 1 year) 4. Signed informed consent from the patient or legal representative.
**Exclusion Criteria**
1. Based on ECG at the Emergency Department: a. Atrial Flutter b. Newly documented Left Bundle Branch Block (LBBB) c. Newly documented Right Bundle Branch Block (RBBB) with QRS duration > 150 ms 2. Previously documented 24 h ECG halter monitoring with >720 polymorphic PVCs/24 h, or non sustained ventricular tachycardia 3. No history of coronary artery disease 4. Acute or within 1 year ST-Segment Elevation Myocardial Infarction (STEMI) 5. Acute or within 1 year Non-ST-Segment Elevation Myocardial Infarction (NSTEMI), according to ESC 2020 guidelines on NSTEMI: a. If troponin at t0 h is over the “low” criterion on table of the cutoff values b. If the change in troponin (Δtroponin) at t1h is over the respective cutoff value at the table for the cutoff values 6. Unstable angina, defined as myocardial ischemia at rest or at minimum effort, in the absence of acute injury/necrosis of myocardial cells 7. Known residual ischemia: a. Positive imaging-based stress testing b. Negative imaging-based stress testing ≥ 1 year, and: ▪ History of known coronary artery stenosis > 60% without revascularization, or ▪ PCI ≥ 1 year, or ▪ CABG ≥ 3 years 8. History of acute coronary syndrome within 1 year 9. Severe Aortic Valve Stenosis (mean pressure gradient > 40 mmHg, AVA < 1 cm/m^2^) 10. Severe Chronic Kidney Disease (stage ≥ 4) 11. Severe systematic disease, including neoplasmatic disease under any antineoplasmatic treatment, liver failure, infection with fever 12. Use of strategy “pill in the pocket”, by taking flecainide (max 200 mg) or propafenone (max 600 mg) within 6 h prior to Emergency Department visit 13. Known dysanexia or allergy to flecainide or amiodarone 14. Pregnancy or/and breastfeeding 15. Participation in any other clinical trial 16. Life expectancy less than 1 year 17. Inappropriate, unfit, or unwilling to follow the designated protocol procedures.

**Table 2 jcm-12-03961-t002:** Summary of type and frequency of safety and efficacy outcomes at each patient’s assessment.

Event	Screening Visit Vscr	Intervention Visit V0	Visit at 24-h (V1)	Visit at 30 Days (V2)
Type of contact	On site	On site	On site	Phone
Study phase	Screening	Intervention	Follow-up	Follow-up
Inclusion/Exclusion criteria	x			
Informed consent	x			
Demographics and medical history	x			
Physical Examination	x	x	x	
Electrocardiogram	x	x	x	
Echocardiography	x (if not within last year)		x	
Chest X-ray	x			
Laboratory Exams	x	x (1 h hs-TnI only)		
24-h ECG holter monitoring		x		
Medication profile	x		x	
Adverse event assessment		x	x	x
Endpoints assessment		x	x	x

**Table 3 jcm-12-03961-t003:** Primary and Secondary Outcomes.

Primary Outcomes
The frequency of successful cardioversion to sinus rhythm [Time Frame: From the drug initiation and for 6 h] The combined frequency of premature ventricular contractions (PVCs), non-sustained ventricular tachycardia (NSVT), sustained ventricular tachycardia (SVT), bradycardia < 50 bpm and systolic blood pressure < 90 mmHg. [Time Frame: From the drug initiation and for 6 h]
**Secondary Outcomes**
The frequency of patient discharges from the Emergency Department in a sinus rhythm [Time Frame: From the drug initiation and for 6 h] The frequency of successful cardioversion to sinus rhythm [Time Frame: From the drug initiation and for 24 h, 24 h ECG Holter monitoring] The time until the cardioversion to sinus rhythm [Time Frame: From the drug initiation and for 6 h] The frequency of electrical cardioversion [Time Frame: From the drug initiation and for 24 h] The frequency of arrhythmias: burden of PVCs, NSVT episodes, SVT episodes [Time Frame: From the drug initiation and for 24 h] The frequency, severity and type of Adverse Events [Time Frame: From the drug initiation and for 30 days]

## Data Availability

Data available on request due to privacy and ethical restrictions.

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
