# Peer review of "Rationale and Design of a Randomized Controlled Clinical Trial on the Safety and Efficacy of Flecainide versus Amiodarone in the Cardioversion of Atrial Fibrillation at the Emergency Department in Patients with Coronary Artery Disease (FLECA-ED)"

_jcm, 2023, doi:10.3390/jcm12123961_

Round 1

Reviewer 1 Report

The authors evaluated the hypothesis that using flecainide for the cardioversion of eligible patients will allow for faster cardioversion periods, while maintaining the same or better safety profile than amiodarone. 

I have the following concerns:

1. Please define the term "successful cardioversion".

2. What kind of centres are enrolled (ertiary hospitals?)?

3. Please explain in details how patients after cardioversion will be monitored?

4. Is the assessment of extracardiac adverse effects of the drug planned?

5. Are patients with hyperthyroidism excluded?

6. What is the EHRA symptom score of the patients?

7. Amiodarone is preferred in patients with structural heart disease. What is the plan of treatment in patients without structural heart disease?

Minor editing of English language required.

Author Response

Dear Reviewer,

I would like to extend my sincere gratitude for the thorough review and constructive comments on our manuscript, "Safety and Efficacy of Flecainide versus Amiodarone in Cardioversion of Atrial Fibrillation".

In response to the comments and suggestions, we have revised our manuscript. Herein, I am detailing the revisions and their locations in the text:

  1. Please define the term "successful cardioversion".
    Response: Definition of "successful cardioversion" has been added (Line 82).

  2. What kind of centres are enrolled (tertiary hospitals?)
    Response: Details about the type of centres enrolled in the study have been provided (Lines 109).

  3. Please explain in details how patients after cardioversion will be monitored?
    Response: Explanation about patient monitoring post-cardioversion has been elaborated (Lines 199, 206-210, 367).

  4. Is the assessment of extracardiac adverse effects of the drug planned?
    Response: Information on the assessment of extracardiac adverse effects of the drug has been added (Lines 251-255).

  5. Are patients with hyperthyroidism excluded?
    Response: Clarification about the eligibility of patients with thyroid conditions is now included (Lines 129-130).

  6. What is the EHRA symptom score of the patients?
    Response: Information on the EHRA symptom score of the patients has been added (Lines 128).

  7. Amiodarone is preferred in patients with structural heart disease. What is the plan of treatment in patients without structural heart disease?
    Response: Future plan of treatment in study patients will be determined according to physician preference (not part of the study protocol).

We hope that these revisions adequately address your comments and improve the quality of our manuscript. We look forward to receiving your continued feedback.

Best Regards,

Dimitris Tsiachris

Reviewer 2 Report

This study is investigating the safety and efficacy of Flecainide versus Amiodarone in the cardioversion of atrial fibrillation at the emergency department.  

1. The author should add abbreviations for example: ED, NOACs, etc.

2. How long is the clinical study ongoing?

3. It would be better if the author provided preliminary data. 

4. The study was designed well, and the protocol was written well.

Author Response

Dear Reviewer,

I would like to extend my sincere gratitude for the thorough review and constructive comments on our manuscript, "Safety and Efficacy of Flecainide versus Amiodarone in Cardioversion of Atrial Fibrillation".

In response to the comments and suggestions, we have revised our manuscript. Herein, I am detailing the revisions and their locations in the text:

  1. The author should add abbreviations for example: ED, NOACs, etc.

Response: Abbreviations have been added (Lines 30-34).

  1. How long is the clinical study ongoing?

Response: Duration of the clinical study has been clarified (Lines 89-90, 382).

  1. It would be better if the author provided preliminary data.

Response: Please note that we are in the process of enrolling patients, and therefore we have no data to report at the present time.

We hope that these revisions adequately address your comments and improve the quality of our manuscript. We look forward to receiving your continued feedback.

Best Regards,

Dimitris Tsiachris

Reviewer 3 Report

The scope of the trial is relevant and properly addresses the long debated issue on the use of flecainide in patients with AF and CAD. The study design is appropriate and thoroughly described. The discussion is comprehensive and takes into account the main studies in the field.

I only suggest to address the following issues before accepting the paper for publication:

Major comments

- It is not clear if the study protocol has been approved from an ethical committee or not. Being a randomized interventional study this is of high relevance. In line 152 it is stated that an Ethical Committee has approved the informed consent, not the full protocol. In lines 107-108 it is stated that "study procedures" are approved by the Hospitals’ Institutional Review Board and the National Organization for Medicines; it is not clear neither if "study procedures" refers to the study protocol nor if the cited entities can be considered equivalent or alternative to an Ethical Committee

- In Table 1, under the "Inclusion Criteria", "History of Coronary Artery Disease without residual ischemia, defined by one of the following criteria" corresponds to the bullet point "c" which, according to the table, is an alternative to the points "a" and "b". In other words, according to this table, a patient with AF onset between 48 hours and 7 days who has not received anticoagulation for at least 30 days but has a history of CAD without residual ischemia could be enrolled into the study. This does not look reasonable and is contradiction with the protocol as described in the text. My impression is that the bullet point "c" should not be an alternative to "a" and "b" but rather a different numerical bullet point (e.g. "3"). If this is the case, it would be appropriate to also correct the corresponding table in clinicaltrials.gov

Minor comments:

- Table 1. It should be clarified if points "4" and "5" refer to history of AMI or ongoing AMI

- Table 1.  What does "poly" mean?

- Table 1. Bullet point "4" before "Signed informed consent from the patient or legal representative." is missing

Author Response

Dear Reviewer,

I would like to extend my sincere gratitude for the thorough review and constructive comments on our manuscript, "Safety and Efficacy of Flecainide versus Amiodarone in Cardioversion of Atrial Fibrillation".

In response to the comments and suggestions, we have revised our manuscript. Herein, I am detailing the revisions and their locations in the text:

  1. It is not clear if the study protocol has been approved from an ethical committee or not. Being a randomized interventional study this is of high relevance. In line 152 it is stated that an Ethical Committee has approved the informed consent, not the full protocol. In lines 107-108 it is stated that "study procedures" are approved by the Hospitals’ Institutional Review Board and the National Organization for Medicines; it is not clear neither if "study procedures" refers to the study protocol nor if the cited entities can be considered equivalent or alternative to an Ethical Committee

    Response:
    Clarification on the approval of study protocol from an ethical committee has been included (Lines 116, 176, 369, 383).

  2. In Table 1, under the "Inclusion Criteria", "History of Coronary Artery Disease without residual ischemia, defined by one of the following criteria" corresponds to the bullet point "c" which, according to the table, is an alternative to the points "a" and "b". In other words, according to this table, a patient with AF onset between 48 hours and 7 days who has not received anticoagulation for at least 30 days but has a history of CAD without residual ischemia could be enrolled into the study. This does not look reasonable and is contradiction with the protocol as described in the text. My impression is that the bullet point "c" should not be an alternative to "a" and "b" but rather a different numerical bullet point (e.g. "3"). If this is the case, it would be appropriate to also correct the corresponding table in clinicaltrials.gov

    Response: We have made necessary corrections in Table 1 regarding inclusion criteria as you correctly pointed out.
  1. It should be clarified if points "4" and "5" refer to history of AMI or ongoing AMI

    Response:
    This has been amended in Table 1.

  2. What does "poly" mean?

    Response: The term "poly" has been expanded to "polymorphic" in Table 1.
  1. Bullet point "4" before "Signed informed consent from the patient or legal representative." is missing
    Response: The missing bullet point has been amended.

We hope that these revisions adequately address your comments and improve the quality of our manuscript. We look forward to receiving your continued feedback.

Best Regards,

Dimitris Tsiachris

Round 2

Reviewer 1 Report

Thank you for your response.

How long should restoration of sinus rhythm last at least to be defined as 'succesful cardioversion?

English language is correct

Author Response

I would like once again to extend my sincere gratitude for the thorough review and constructive comments on our manuscript. This specific comment is indeed important. We have defined “ successful cardioversion” as the restoration of sinus rhythm, documented by 12-lead ECG, lasting at least 1 hour (line 83).

Reviewer 2 Report

This study is investigating the safety and efficacy of Flecainide versus Amiodarone in the cardioversion of atrial fibrillation at the emergency department.  

The manuscript is written well.

Author Response

I would like once again to extend my sincere gratitude for the thorough review and constructive comments on our manuscript.